# GENERATIVE PROMPTING WITH DIFFUSION FOR LIFE-LONG CONTINUAL ADAPTATION

## ABSTRACT

Machine learning models deployed in dynamic environments often face distribution shifts that are not entirely novel but instead recurring and long-term. To capture this practical scenario, we introduce Lifelong Continual Adaptation (LCA), where models are trained on multiple domains and then deployed in sequential streams in which these domains recur over time. Because such recurrences can be anticipated, LCA seeks to reuse domain-specific knowledge without retraining whenever a domain reappears. While similar in using sequential test streams, continual Test-Time Adaptation (TTA) assumes each domain is unseen, i.e., out-of-distribution (OOD). Applied to LCA, its reliance on online unsupervised training is dispensable (no novel domains to relearn), unstable (errors accumulate across recurrences), and inefficient (due to costly backpropagation on large models). To overcome these issues, we propose DiffPrompt, a diffusion-based prompt generation framework that produces domain-specific prompts to guide a frozen vision foundation model. A conditional diffusion model learns the distribution of prompts across domains during training and generates prompts conditioned on incoming data batches during deployment. Experiments on DomainNet and ImageNet-C show that DiffPrompt achieves stable and efficient adaptation, outperforming ERM and continual TTA baselines and validating LCA as a realistic and non-trivial setting.

## 1 INTRODUCTION

Modern machine learning models must operate in dynamic environments where data distributions evolve over time. In many real-world deployments, these shifts are not entirely novel but instead recurring and long-term. For example, autonomous driving systems repeatedly encounter similar conditions across day–night cycles and seasonal weather changes; mobile vision systems face recurring app usage patterns; and surveillance models adapt to periodic lighting and crowd variations. These scenarios highlight the need for models to retain and efficiently reuse domain knowledge whenever a previously encountered distribution reappears.

To address such practical scenarios, we introduce Lifelong Continual Adaptation (LCA). In LCA, a model is trained on data from multiple domains and then deployed in a sequential stream where these domains recur over time, as illustrated in Figure 1. Since such recurrences can be anticipated, LCA assumes that training samples from these domains are available before deployment. The central challenge is to leverage this prior knowledge for stable and efficient adaptation during deployment, without retraining the model each time a domain reappears. LCA thus captures a realistic property of deployment data streams: domains are not always unseen, but instead frequently recur.

Existing research settings only partially reflect this reality. Continual learning primarily addresses catastrophic forgetting when learning new tasks sequentially (Wang et al., 2024b), but it does not model the ability to efficiently recall and adapt to recurring domains at test time. The most closely related setting is continual Test-Time Adaptation (TTA) (Wang et al., 2022), which also models sequential domains but assumes that each domain is out-of-distribution (OOD), during deployment. By contrast, LCA focuses on in-distribution (ID) recurring domains—all domains are observed during training, yet the model must repeatedly adapt to unseen test samples from them during deployment. This difference exposes a fundamental mismatch between continual TTA and the needs of LCA: optimization-based TTA methods are tailored for unseen OOD shifts, relying on online un-

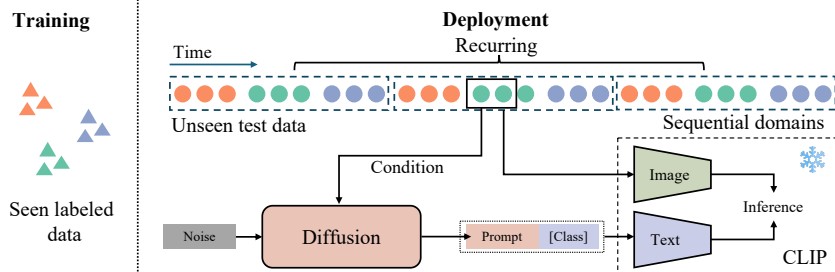

Figure 1: This work studies scenarios where a model operates in data streams with sequential and recurring domains. Different colors represent different domains. The deployment part illustrates the adaptation workflow of the proposed DiffPrompt method, where a diffusion model generates domain-specific prompts conditioned on incoming data to guide a frozen foundation model.

supervised training. Under the ID recurring-domain scenario of LCA, however, such training is **dispensable** (since no novel domain needs to be relearned), **unstable** (due to error accumulation across recurrences as shown in Table 3), and **inefficient** (due to the high cost of backpropagation in large foundation models).

To address these issues, we propose DiffPrompt. We take inspiration from the prompt tuning paradigm (Zhou et al., 2022b; Jia et al., 2022), which adapts foundation models by optimizing only the learnable prompt vectors while keeping the backbone frozen. Building on this idea, we design a diffusion-based prompt generation framework that enables vision foundation models, such as CLIP (Radford et al., 2021), to adapt to long-term sequential domains under resource constraints. Specifically, DiffPrompt employs a diffusion model as a domain prompt generator, directly sampling prompts through an iterative denoising process conditioned on incoming batches of data. Training proceeds in two stages: prompt collection and diffusion training. The first stage encapsulates each domain's distribution through a set of learnable prompt samples, where domain-specific knowledge is encoded. The second stage trains a conditional latent diffusion model on these prompt samples, enabling the generation of domain-specific prompts from Gaussian noise conditioned on image batches during deployment (Figure 1). This generative approach offers two key advantages: (1) it models a distribution of prompts rather than relying on a single one, improving robustness; and (2) it eliminates the need for gradient backpropagation through the backbone, achieving significant resource efficiency.

We evaluate DiffPrompt against multiple baseline methods under the proposed LCA setting, including the non-adapted Empirical Risk Minimization (ERM) (Vapnik et al., 1998) and several continual TTA methods (Wang et al., 2022; Döbler et al., 2023; Yuan et al., 2023). Results not only demonstrate the effectiveness of DiffPrompt but also validate that LCA is a non-trivial and realistic deployment scenario—one that existing settings do not adequately capture.

Overall, our work makes three contributions. First, we introduce a novel problem setting, Lifelong Continual Adaptation, which emphasizes the evaluation of long-term sequential, recurring, and in-distribution domains in real-world scenarios. Second, we propose a framework that formulates the continual adaptation process as generating domain-specific prompts via a diffusion-based generative approach. Diffusion models have been successfully applied to generating inputs (e.g., images (He et al., 2023)), outputs (e.g., bounding boxes (Chen et al., 2023), semantic labels (Tan et al., 2022)), and neural network parameters (Erkoç et al., 2023). Our work is the first to show that diffusion models can also be used to generate prompts for continual adaptation. Finally, we conduct extensive experiments to demonstrate the effectiveness of the proposed method compared to various baselines.

## 2 RELATED WORK

**Prompt tuning and adaptation.** Prompting (Petroni et al., 2019; Brown et al., 2020; Lester et al., 2021; Li & Liang, 2021; Liu et al., 2021; Yao et al., 2023; Zhu et al., 2023) has emerged as a crucial technique for enhancing the performance of pre-trained models in various downstream tasks. Radford et al. (2021) introduces CLIP, a powerful vision-language model which uses textual prompts

to guide image classification. Following this, CoOp (Zhou et al., 2022b) proposes to adapt CLIP by learning textual prompts on the text encoder. CoCoOp (Zhou et al., 2022a) extends CoOp by conditionally tuning to improve performance. VPT (Jia et al., 2022) introduces fine-tuning prompts on Vision Transformers (Dosovitskiy et al., 2021) to adapt to downstream tasks. Their work explores transferring foundation models to specific tasks and domains through prompt tuning. This paradigm involves tuning only a small number of parameters associated with the input, delivering efficiency and preserving the pre-trained backbone models.

**Continual adaptation.** Some work (Hoffman et al., 2014; Wulfmeier et al., 2018; Volpi et al., 2021; Liu et al., 2020; Kumar et al., 2020) considers an evolving domain adaptation where the target domain evolves over time. A line of recent research known as continual Test-Time Adaptation (TTA) focuses on continually adapting a source model to target unseen sequential domains (Wang et al., 2022). These methods are mainly based on a teacher-student self-training framework and utilize source prototype pulling (Döbler et al., 2023) and resampling (Yuan et al., 2023) strategies to improve stability. The LCA setting in our work differs from continual TTA in its focus. Continual TTA aims to prevent performance degradation across sequential domain shifts by learning from unseen domains during test time. In contrast, LCA focuses on achieving stable and high performance by retrieving knowledge for each encountered seen domain during deployment.

**Continual learning.** Continual learning addresses catastrophic forgetting, the performance degradation on old tasks when learning new ones (Wang et al., 2024b). Domain incremental learning, a subset of continual learning, involves sequential domains and aims to balance performance across old and new domains (Mirza et al., 2022; Shi & Wang, 2023). In contrast, LCA focuses on retrieving and utilizing knowledge of an old domain during test time, without learning new domains.

**Diffusion-based generation.** Denoising Diffusion Probabilistic Models (DDPM) have garnered significant attention for their ability to produce high-quality data through a process of iterative denoising (Ho et al., 2020; Luo, 2022; Rombach et al., 2022; Dhariwal & Nichol, 2021; Peebles & Xie, 2023; Croitoru et al., 2023). Several studies employ diffusion models for data augmentation (He et al., 2023; Trabucco et al., 2024), and these models are also explored for classification tasks (Li et al., 2023; Du et al., 2023; Prabhudesai et al., 2023). Furthermore, recent research investigates the application of diffusion models for generating neural network weights (Erkoç et al., 2023; Nava et al., 2023; Wang et al., 2024a). Some work also utilizes diffusion models to generate bounding boxes for object detection (Chen et al., 2023) and to enhance the quality of semantic segmentation (Tan et al., 2022).

# 3 METHODOLOGY

We propose a diffusion-based prompt generation method to achieve stable and high-performance deployment of a foundation model on practical data streams involving sequential and recurring domains. Section 3.1 introduces the preliminaries of the problem definition and diffusion models. Section 3.2 describes the training procedure before deployment. Section 3.3 introduces the diffusion condition module. Finally, Section 3.4 covers the deployment of our method.

## 3.1 PRELIMINARY

### 3.1.1 SETTING OF LIFELONG CONTINUAL ADAPTATION

Let $\mathcal{D}_1, \mathcal{D}_2, \ldots, \mathcal{D}_n$ represent different domains, each with a corresponding training set $\mathcal{D}_i^{\text{train}}$ and a test set $\mathcal{D}_i^{\text{test}}$. During **training**, the model has access to the training sets $\{\mathcal{D}_i^{\text{train}}\}_{i=1}^n$ from these multiple domains. During **deployment**, the model encounters test sets from these domains in a sequential and recurring manner. Let $\mathcal{S} = \{\mathcal{D}_1, \mathcal{D}_2, \ldots, \mathcal{D}_n\}$ represent the sequence of domains. The test stream presents this sequence of domains, which recurs $r$ times, as represented:

$$S^{test} = \{(D_1^{test}, D_2^{test}, \ldots, D_n^{test})_1, (D_1^{test}, D_2^{test}, \ldots, D_n^{test})_2, \ldots, (D_1^{test}, D_2^{test}, \ldots, D_n^{test})_r\} \tag{1}$$

Specifically, the model performs adaptation and inference on the data examples from each domain $D_i^{\text{test}}$. We use the classification accuracy $A_{i,j}$ as the evaluation metric, corresponding to the performance on the $i$-th domain during the $j$-th recurrence. In addition, we calculate the mean accuracy

Figure 2: Overview of training. The training of DiffPrompt involves two stages: prompt collection (left) and diffusion training (right). In the prompt collection stage, a base model with trainable prompts is trained using the training sets from different domains to collect prompt samples. In diffusion training, these collected prompt samples are used to train a conditional latent diffusion model for prompt generation at deployment time. The deployment process is illustrated in Figure 1.

over the entire data stream as follow:

$$\bar{A} = \frac{1}{nr} \sum_{j=1}^{r} \sum_{i=1}^{n} A_{i,j}. \tag{2}$$

This setting reflects the practical deployment of machine learning models. A deployed model typically handles data streams involving sequential and recurring domains over time. For example, autonomous driving systems process visual data across different weather conditions and throughout day-night cycles. Before deployment, we can collect data samples from these domains and learn relevant knowledge, enabling stable and high-performance deployments.

### 3.1.2 DIFFUSION MODELS

Diffusion models are a sophisticated class of generative models that have shown remarkable capabilities in generating high-quality synthetic data. The core principle behind diffusion models involves a process known as the forward and reverse diffusion processes (Ho et al., 2020; Luo, 2022; Rombach et al., 2022).

**Forward diffusion.** The original data is gradually noised in forward diffusion. Specifically, for data $x_0$ sampled from the real distribution $q(x)$, the forward diffusion $q(x_{1:T}|x_0)$ is a process of adding noise to the data with a Markov chain of $T$ steps of $q(x_t|x_{t-1})$, at each of which Gaussian noise with variance $\beta_t$ is added:

$$q(x_{1:T}|x_0) = \prod_{t=1}^{T} q(x_t|x_{t-1}), \text{ where } q(x_t|x_{t-1}) = \mathcal{N}(x_t; \mu_t = \sqrt{1-\beta_t}x_{t-1}, \Sigma_t = \beta_t I), \tag{3}$$

where $\mu$ and $\Sigma$ are the mean and variance, and $I$ is the identity matrix.

**Reverse diffusion.** The forward process adds noise incrementally until $x_T$ resembles isotropic Gaussian noise. Consequently, we can sample a $x_T$ from a Gaussian distribution $\mathcal{N}(0, I)$ and conduct a reverse diffusion to generate a sample $x \sim q(x)$. Because $q(x_{t-1}|x_t)$ is intractable to compute, we use a neural network $p_\theta$ parameterized with $\theta$ to approximate it:

$$p_\theta(x_{0:T}) = p_\theta(x_T) \prod_{t=1}^{T} p_\theta(x_{t-1}|x_t), \text{ where } p_\theta(x_{t-1}|x_t) = \mathcal{N}(x_{t-1}; \mu_\theta(x_t, t), \Sigma_\theta(x_t, t)), \tag{4}$$

where $\boldsymbol{\mu}_\theta$ and $\boldsymbol{\Sigma}_\theta$ are the predicted Gaussian parameters by the diffusion model.

The training of the diffusion model involves the optimization of the negative log-likelihood of the training data. A simplified version of the evidence lower bound is typically used as the objective function:

$$\mathcal{L} = \mathbb{E}_{\boldsymbol{x}_0, t, \boldsymbol{\epsilon} \sim \mathcal{N}(0, \boldsymbol{I})} \left[ \| \boldsymbol{\epsilon} - \boldsymbol{\epsilon}_\theta(\sqrt{\bar{\alpha}_t}\boldsymbol{x}_0 + \sqrt{1 - \bar{\alpha}_t}\boldsymbol{\epsilon}, t) \|^2 \right], \tag{5}$$

where $\alpha_t = 1 - \beta_t$ and $\bar{\alpha}_t = \prod_{s=1}^{t} \alpha_s$; $\boldsymbol{\epsilon}_\theta$ is the neural network used to predict the noise $\boldsymbol{\epsilon}$ at each time step $t$.

## 3.2 OVERVIEW OF TRAINING

There are two stages during the training of DiffPrompt: prompt collection and diffusion training, as illustrated in Figure 2. In the prompt collection stage, we train a base model with trainable prompts using the training sets of different domains to collect prompt samples. In diffusion training, we train a diffusion model with the collected prompt samples for prompt generation at deployment time.

**Prompt collection.** We employ the foundation model CLIP (Radford et al., 2021) as the base model. Upon it, we adopt trainable textual prompts (Zhou et al., 2022b) to realize adaptation for the model. For each domain in $\{\mathcal{D}_i^{\text{train}}\}_{i=1}^{n}$, we train the base model using the cross-entropy loss:

$$\mathcal{L}_{\text{CE}} = -\log \frac{\exp(\cos(\boldsymbol{F}, \boldsymbol{T}_y)/\tau)}{\sum_{j=1}^{C} \exp(\cos(\boldsymbol{F}, \boldsymbol{T}_j)/\tau)}, \tag{6}$$

where $\boldsymbol{F}$ is the image encoder output; $\boldsymbol{T}_j$ is the text encoder output for the $j$-th class out of $C$ classes; $\cos(\cdot, \cdot)$ denotes the cosine similarity, and $\tau$ is a temperature parameter. Only the prompt is tunable while the whole CLIP model is frozen. For each domain, we collect a set of fitted prompts from different epochs, expressed as:

$$\mathcal{P} = \{\mathcal{P}_k \mid k = 1 \sim n\}, \text{ where } \mathcal{P}_k = \{\boldsymbol{p}_{k,j} \mid j = 1 \sim m\}, \tag{7}$$

where $m$ is the number of prompt samples for each domain.

**Diffusion training.** After collecting the prompt samples, we train a conditional latent diffusion model with them to learn the prompt-space distribution among different domains. We first train an unconditional denoising autoencoder $f_{ae}$ to translate the prompts $\mathcal{P}$ to a low-dimensional latent space. It is optimized using a reconstruction loss:

$$\mathcal{L}_{ae} = \frac{1}{B} \sum_{i=1}^{B} \| \boldsymbol{p}_i - f_{ae}(\boldsymbol{p}_i, \boldsymbol{\xi}) \|^2 \tag{8}$$

where $\boldsymbol{p}_i$ represents the $i$-th prompt in a batch; $\boldsymbol{\xi}$ is the random noise added to the input and the latent space; $f_{ae}(\boldsymbol{p}_i, \boldsymbol{\xi})$ is the reconstructed prompt from the autoencoder with added noise, and $B$ is the batch size. The latent space aids in efficient and stable training (Rombach et al., 2022).

Under the latent space, we train an image-conditioned diffusion model using the prompt samples and training images. This process can be expressed as:

$$\theta = \theta - \gamma \nabla_\theta \left\| \boldsymbol{\epsilon} - \boldsymbol{\epsilon}_\theta \left( \sqrt{\bar{\alpha}_t}\boldsymbol{v}_0 + \sqrt{1 - \bar{\alpha}_t}\boldsymbol{\epsilon}, t, \mathcal{C}(\boldsymbol{X}^{\text{train}}) \right) \right\|^2. \tag{9}$$

Here, $\boldsymbol{v}_0$ is the latent representation of the prompts, and $\mathcal{C}$ is the designed condition module used to encode images into conditions, which will be described in Section 3.3. $\boldsymbol{X}^{\text{train}}$ represents the training images from the corresponding domain associated with $\boldsymbol{v}_0$. $\gamma$ is the learning rate.

## 3.3 FEATURE DISTRIBUTION AS CONDITION

The condition module is designed to perceive domains in order to provide conditions that are sensitive to different domains. Inspired by discussions on the relationship between domains and feature distribution in the field of domain adaptation, we recognize that domain shifts result in varied feature distributions extracted by discriminative models. A line of research focuses on overcoming this issue by forcing models to extract similar feature distributions from images of different domains to

achieve domain adaptation (Ganin et al., 2016; Ganin & Lempitsky, 2015; Sun & Saenko, 2016). In this work, however, we exploit this characteristic within the condition module; in other words, we use feature distributions as conditions.

Concretely, we employ a pre-trained image encoder from CLIP in the condition module. It produces features $\boldsymbol{F}$ from the incoming batch of images from streams. Afterward, we compute the distribution statistics for the batch of features, adopting the mean and standard deviation as follows:

$$\boldsymbol{\mu} = \frac{1}{c} \sum_{i=1}^{c} \boldsymbol{F}_i, \quad \boldsymbol{\sigma} = \sqrt{\frac{1}{c} \sum_{i=1}^{c} (\boldsymbol{F}_i - \boldsymbol{\mu})^2}, \tag{10}$$

where $c$ is the batch size of images. We concatenate the mean and standard deviation as $\text{concat}[\boldsymbol{\mu}, \boldsymbol{\sigma}]$ to serve as the conditions to be input to the diffusion model.

### 3.4 IMAGE-CONDITIONED PROMPT GENERATION

At deployment time, as shown in Figure 1, we sample prompts conditioned on batches of test images from test streams using the trained diffusion model through the reverse diffusion process:

$$\boldsymbol{v}_{t-1} = \frac{1}{\sqrt{\alpha_t}} \left( \boldsymbol{v}_t - \frac{1 - \alpha_t}{\sqrt{1 - \bar{\alpha}_t}} \boldsymbol{\epsilon}_\theta(\boldsymbol{v}_t, t, \mathcal{C}(\boldsymbol{X}^{\text{test}})) \right) + \sigma_t \boldsymbol{z}, \tag{11}$$

where $t = T, \ldots, 1$; $\boldsymbol{v}_T$ and $\boldsymbol{z}$ are random noise sampled from $\mathcal{N}(0, \boldsymbol{I})$; $\sigma_t$ comes from the noise schedule used in the forward diffusion process, and $\boldsymbol{X}^{\text{test}}$ represents the test images. The generated prompts are assigned to the base model to adapt it to the current domain.

## 4 EXPERIMENTS

### 4.1 DATASETS

We include two datasets in the experiments, DomainNet (Peng et al., 2019) and ImageNet-C (Croce et al., 2021), which are widely used in domain adaptation and test-time adaptation tasks.

**DomainNet** The dataset contains 6 domains: Clipart (clip), Infograph (info), Painting (paint), Quickdraw (quick), Real (real), and Sketch (sketch). It has 345 classes and is naturally class-imbalanced in each domain. We follow the official split to organize the training and test sets.

**ImageNet-C** This dataset is derived by applying various corruptions to the images in the validation set of ImageNet. There are 4 categories of corruptions (weather, noise, blur, digital) aimed at mimicking a range of natural environmental conditions that may be encountered during deployment. Following the RobustBench benchmark (Croce et al., 2021), we adopt 15 corruption domains, including brightness (bri), frosted glass (gla), JPEG compression (jpe), contrast (con), defocus blur (def), impulse noise (imp), motion blur (mot), snow (sno), zoom blur (zoo), frost (fro), pixelation (pix), gaussian noise (gau), elastic transformation (ela), shot noise (sho), and fog (fog). Similar to DomainNet, we adopt a $70\%/30\%$ split to obtain the training and test sets for each domain.

### 4.2 EXPERIMENTAL SETUP

The experimental setting mimics a practical scenario where a model is deployed in data streams involving sequential and recurring domains. It has two primary conditions. Firstly, a sequence of domains is presented in streams, which is the same setup as recent continual TTA works (Song et al., 2023; Döbler et al., 2023), with associated experimental results detailed in Tables 1 and 2. Secondly, the sequence of domains in the first condition recurs in streams with recurring times set to 15, and the associated results are shown in Table 3. For all methods, we use the same base model, which is the 'ViT-B/16' version of the CLIP model. All methods follow the paradigm of prompt tuning, where only the prompt is updated while CLIP's model parameters remain frozen. The batch size is uniformly set to 64. TTA methods are evaluated with the ERM-trained model, ensuring that all domains are seen.

Table 1: Continual adaptation on test sets of 6 sequential domains in DomainNet. Results report accuracy (± std over seeds). TTA methods use the ERM-trained model for fair comparison.

| Method | clip | info | paint | quick | real | sketch | Average |
|---|---|---|---|---|---|---|---|
| Zero-shot | 71.0 | 47.6 | 66.2 | 13.9 | 83.7 | 63.5 | 57.7 |
| ERM | 75.3 | 55.6 | 72.3 | 25.5 | 85.9 | 68.0 | 63.8 |
| TPT | 76.5 | 59.4 | 73.9 | 24.0 | 86.4 | 69.4 | 64.9 |
| TENT | 75.4 $\pm 0.01$ | 53.6 $\pm 0.02$ | 69.5 $\pm 0.04$ | 1.8 $\pm 0.05$ | 84.4 $\pm 0.02$ | 50.4 $\pm 1.80$ | 55.9 $\pm 0.30$ |
| SAR | 75.0 $\pm 0.03$ | 53.5 $\pm 0.09$ | 69.8 $\pm 0.07$ | 11.7 $\pm 0.84$ | 85.0 $\pm 0.17$ | 65.1 $\pm 0.40$ | 60.0 $\pm 0.19$ |
| CoTTA | 75.3 $\pm 0.03$ | 55.4 $\pm 0.02$ | 72.2 $\pm 0.06$ | 23.4 $\pm 0.58$ | 85.4 $\pm 0.28$ | 67.2 $\pm 0.37$ | 63.2 $\pm 0.22$ |
| RMT | 74.7 $\pm 0.18$ | 55.0 $\pm 0.15$ | 70.8 $\pm 0.35$ | 20.0 $\pm 1.76$ | 85.9 $\pm 0.16$ | 67.6 $\pm 0.41$ | 62.4 $\pm 0.48$ |
| RoTTA | 75.4 $\pm 0.05$ | 55.5 $\pm 0.05$ | 72.1 $\pm 0.08$ | 22.6 $\pm 1.24$ | 85.5 $\pm 0.22$ | 67.5 $\pm 0.30$ | 63.1 $\pm 0.30$ |
| Ours | **79.6** $\pm 0.06$ | **58.9** $\pm 0.23$ | **75.8** $\pm 0.69$ | **30.1** $\pm 0.14$ | **87.7** $\pm 0.48$ | **71.5** $\pm 0.50$ | **67.3** $\pm 0.29$ |

Table 2: Continual adaptation on test sets from 15 sequential domains in ImageNet-C. TTA methods are evaluated using the ERM-trained model. Results report accuracy (± std over seeds).

| Method | bri | gla | jpe | con | def | imp | mot | sno | zoo | fro | pix | gau | ela | sho | fog | Avg |
|---|---|---|---|---|---|---|---|---|---|---|---|---|---|---|---|---|
| Zero-shot | 53.0 | 15.1 | 32.1 | 21.6 | 23.1 | 14.6 | 24.1 | 28.1 | 22.0 | 28.5 | 32.3 | 15.2 | 13.2 | 15.7 | 38.1 | 25.1 |
| ERM | 57.6 | 19.9 | 36.7 | 26.7 | 27.7 | 20.2 | 29.4 | 33.4 | 26.8 | 33.4 | 38.4 | 20.3 | 18.6 | 20.2 | 43.2 | 30.2 |
| TENT | 46.7 $\pm 0.04$ | 5.5 $\pm 0.23$ | 0.6 $\pm 0.01$ | 0.2 $\pm 0.02$ | 0.1 $\pm 0.00$ | 0.2 $\pm 0.01$ | 0.1 $\pm 0.00$ | 0.1 $\pm 0.00$ | 0.1 $\pm 0.01$ | 0.2 $\pm 0.02$ | 0.1 $\pm 0.00$ | 0.1 $\pm 0.00$ | 0.1 $\pm 0.00$ | 0.1 $\pm 0.00$ | 0.1 $\pm 0.01$ | 3.6 $\pm 0.01$ |
| CoTTA | 46.1 $\pm 0.04$ | 12.9 $\pm 0.03$ | 28.2 $\pm 0.06$ | 19.7 $\pm 0.07$ | 19.9 $\pm 0.10$ | 12.7 $\pm 0.07$ | 20.4 $\pm 0.10$ | 23.7 $\pm 0.10$ | 18.0 $\pm 0.21$ | 23.8 $\pm 0.23$ | 26.7 $\pm 0.34$ | 12.5 $\pm 0.16$ | 10.1 $\pm 0.12$ | 12.6 $\pm 0.39$ | 31.4 $\pm 0.19$ | 21.3 $\pm 0.13$ |
| RoTTA | 46.2 $\pm 0.03$ | 13.2 $\pm 0.18$ | 28.9 $\pm 0.28$ | 20.5 $\pm 0.45$ | 20.9 $\pm 0.41$ | 13.8 $\pm 0.47$ | 20.9 $\pm 0.22$ | 24.3 $\pm 0.26$ | 18.6 $\pm 0.14$ | 23.8 $\pm 0.25$ | 26.7 $\pm 0.35$ | 12.7 $\pm 0.15$ | 10.4 $\pm 0.06$ | 12.7 $\pm 0.40$ | 30.0 $\pm 0.90$ | 21.6 $\pm 0.07$ |
| Ours | **60.8** $\pm 0.00$ | **21.0** $\pm 0.00$ | **38.4** $\pm 0.05$ | **27.9** $\pm 0.05$ | **28.3** $\pm 0.00$ | **20.9** $\pm 0.05$ | **30.5** $\pm 0.05$ | **35.4** $\pm 0.04$ | **28.3** $\pm 0.00$ | **34.6** $\pm 0.05$ | **39.4** $\pm 0.05$ | **20.4** $\pm 0.00$ | **22.0** $\pm 0.04$ | **20.8** $\pm 0.05$ | **45.5** $\pm 0.00$ | **31.6** $\pm 0.01$ |

### 4.3 BASELINES

We include eight baseline methods in the comparative experiments: Zero-shot, Empirical Risk Minimization (ERM), two fully test-time adaptation methods—TENT (Wang et al., 2021) and SAR (Niu et al., 2023)—three recent continual test-time adaptation methods—RMT (Döbler et al., 2023), CoTTA (Wang et al., 2022), and RoTTA (Yuan et al., 2023)—as well as a test-time prompt tuning method for CLIP, TPT (Shu et al., 2022). Please refer to Appendix B for detailed information and baseline setups.

### 4.4 IMPLEMENTATION DETAILS

In the prompt collection stage, we tune the prompts for 100 epochs on the training sets of each domain, respectively. Similar to (Wang et al., 2024a), we collect one prompt sample per epoch during the last 80 epochs, resulting in a total of 80 prompt samples per domain. We use the SGD optimizer with a learning rate of 0.003, a momentum of 0.9, and a weight decay of 0.0003. To prevent overfitting, a cosine learning rate scheduler with a warmup period of 2 epochs is applied.

In the diffusion training stage, we follow the autoencoder architecture in (Wang et al., 2024a) for the latent space. For the diffusion model, we adopt an architecture similar to Stable Diffusion (Rombach et al., 2022), but we reduce the model size considering the prompt data scale and add linear layers at both input and output to fit the data to a 1D latent space. We use a linear Beta scheduler with $\beta_{\text{start}} = 0.0001$, $\beta_{\text{end}} = 0.02$, and 1000 steps. For this stage, we use the AdamW optimizer (Loshchilov & Hutter, 2019) with a weight decay of 2e-6 and a learning rate of 0.003. Both stages can be performed on a single 20GB GPU. The Code will be available online.

### 4.5 RESULTS

**On sequential domains.** Tables 1 and 2 present the results of continual adaptation on sequential domains for DomainNet and ImageNet, respectively. We perform multiple runs with five different random seeds for the evaluation. In Table 1, TENT and SAR degrade the performance of the ERM-trained model in the stream under the prompt-tuning paradigm. Regarding continual TTA methods, CoTTA and RoTTA retain stable performance, aligning with the ERM-trained model. Although

Table 3: Continual adaptation on a recurring 6-domain sequence in DomainNet and a recurring 15-domain sequence in ImageNet-C, each repeated 15 times. TTA methods are evaluated using the ERM-trained model.

| | Method | 1 | 2 | 3 | 4 | 5 | 6 | 7 | 8 | 9 | 10 | 11 | 12 | 13 | 14 | 15 | Mean |
|---|---|---|---|---|---|---|---|---|---|---|---|---|---|---|---|---|---|
| **DomainNet** | Zero-shot | 57.7 | 57.7 | 57.7 | 57.7 | 57.7 | 57.7 | 57.7 | 57.7 | 57.7 | 57.7 | 57.7 | 57.7 | 57.7 | 57.7 | 57.7 | 57.7 |
| | ERM | 63.8 | 63.8 | 63.8 | 63.8 | 63.8 | 63.8 | 63.8 | 63.8 | 63.8 | 63.8 | 63.8 | 63.8 | 63.8 | 63.8 | 63.8 | 63.8 |
| | CoTTA | 63.6 | 63.0 | 62.5 | 62.0 | 61.7 | 61.4 | 61.2 | 61.0 | 60.8 | 60.6 | 60.4 | 60.3 | 60.1 | 60.0 | 59.8 | 61.2 |
| | RMT | 61.4 | 61.9 | 62.2 | 62.4 | 62.5 | 62.6 | 62.7 | 62.7 | 62.8 | 62.8 | 62.9 | 62.9 | 62.9 | 62.9 | 62.9 | 62.6 |
| | RoTTA | 63.7 | 63.0 | 62.1 | 60.9 | 59.5 | 57.6 | 55.5 | 53.3 | 50.9 | 48.5 | 46.2 | 44.0 | 42.2 | 40.6 | 39.1 | 52.5 |
| | **Ours** | **67.2** | **67.1** | **67.1** | **66.9** | **67.2** | **67.1** | **67.0** | **67.2** | **66.9** | **67.1** | **67.3** | **67.3** | **67.1** | **67.2** | **67.2** | **67.1** |
| **ImgNet-C** | Zero-shot | 25.1 | 25.1 | 25.1 | 25.1 | 25.1 | 25.1 | 25.1 | 25.1 | 25.1 | 25.1 | 25.1 | 25.1 | 25.1 | 25.1 | 25.1 | 25.1 |
| | ERM | 30.2 | 30.2 | 30.2 | 30.2 | 30.2 | 30.2 | 30.2 | 30.2 | 30.2 | 30.2 | 30.2 | 30.2 | 30.2 | 30.2 | 30.2 | 30.2 |
| | CoTTA | 21.5 | 21.3 | 20.7 | 20.0 | 19.1 | 18.3 | 17.5 | 16.8 | 16.1 | 15.4 | 14.8 | 14.2 | 13.7 | 13.2 | 12.7 | 17.0 |
| | RoTTA | 21.6 | 21.4 | 19.9 | 18.6 | 17.7 | 16.9 | 16.3 | 15.7 | 15.2 | 14.7 | 14.3 | 13.9 | 13.6 | 13.2 | 12.9 | 16.4 |
| | **Ours** | **31.6** | **31.6** | **31.5** | **31.6** | **31.6** | **31.6** | **31.6** | **31.6** | **31.6** | **31.6** | **31.6** | **31.6** | **31.6** | **31.6** | **31.6** | **31.6** |

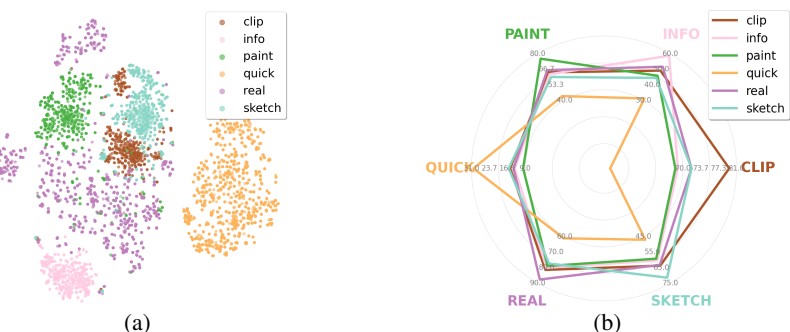

(a)          (b)

Figure 3: (a) Visualization of the conditions produced by the condition module. (b) Performance of generative prompts conditioned on images from different domains. Each color denotes a conditioning domain; prompts perform best when conditioned on the same domain as the test data.

these methods can improve performance from the source to the target domain by learning from test-time data of unseen domains, they do not further enhance performance on ID sequential domains beyond the ERM-trained model. Continual TTA methods focus on learning from unseen domains to prevent model degradation while overlooking performance gains in in-distribution deployment. In contrast, our method demonstrates better deployment performance than the ERM baseline by 3.5%. In Table 2, the continual TTA methods fail to maintain performance with the base model, while our generative expert prompts result in a 1.4% improvement over the ERM-trained prompt.

Unlike continual TTA baselines, which update prompts continuously along the data stream—leading to potential error accumulation—our approach generates separate prompts at each time step, ensuring stable adaptation. The effectiveness of our method depends on the severity of distribution shifts. For ImageNet-C, composed of 15 corruption types (e.g., brightness, contrast, defocus blur), the overall improvement is modest (+1.4%), but larger gains are observed in certain corruptions, such as brightness (+3.2%), elastic transformation (+3.4%), and fog (+2.3%), due to significant pixel-level variation. DomainNet presents more substantial shifts across domains with different image styles, resulting in a stronger average improvement.

**On long-term recurring streams.** Table 3 present results of continual adaptation on long-term sequential and recurring domains for DomainNet and ImageNet, respectively. Here, the domain sequence recurs 15 times. Since each domain is repeated multiple times with different random seeds, we conduct the long-term evaluation only once. CoTTA and RoTTA lead to gradually degraded performance along the recurring episodes. RMT presents stable accuracy approaching that of the ERM-trained model, benefiting from the training prototype pulling. Meanwhile, our method showcases stability with the recurring domain sequence and performs better than the ERM baseline.

## 4.6 ABLATION STUDIES

**Does the condition module perceive different domains?** The condition module is designed to perceive different domains so that it can produce conditions from images to guide the diffusion process

in a domain-aware manner. To verify this, we visualize the output conditions of the condition module fed with test images from different domains. As shown in the t-SNE visualization in Figure 3a, the conditions from test images of different domains present a visually distinguishable distribution, which proves the domain-aware ability of the condition module. Furthermore, the overlapping areas between clusters of different colors, such as Sketch (cyan) and Clip (brown), indicate that these domains share similarities. To verify the similarity, Figure 3b shows that when generated prompts from different domains are applied to each test domain, the prompt conditioned on Sketch images demonstrates the second-best performance on the Clip domain, and vice versa.

**Are the image-conditioned generative prompts domain-specific?** The generative prompts conditioned on the incoming batch of images are expected to be domain-specific and tailored to the current domain. To verify this, we conduct an inter-domain-condition experiment. Specifically, we test the generative prompts conditioned on images from one domain across the sequential domain streams. As shown in Figure 3b, each colored line represents a different domain of images that the prompt generation is conditioned on. The names surrounding the circles indicate the test domains. It can be observed that, for each test domain, the generative prompts conditioned on images from the same domain exhibit the best performance compared to those conditioned on images from other domains. This demonstrates that the generated prompts are domain-specific and well-suited for the encountered domain.

**Study on alternatives to the diffusion model.** We design two alternatives to the diffusion model to validate its effectiveness in our framework. First, we replace the diffusion model with a custom hypernetwork baseline (Ha et al., 2017). Specifically, the hypernetwork uses CLIP's image encoder as the backbone and a linear layer to generate prompts.

Table 4: Study on alternatives to the diffusion model.

| Method | clip | info | paint | quick | real | sketch | Mean |
|---|---|---|---|---|---|---|---|
| Hypernetwork | 65.7 | 46.6 | 59.9 | 12.8 | 77.2 | 60.1 | 53.7 |
| ERM for each | 76.5 | 57.7 | 72.9 | **30.2** | 87.1 | 70.2 | 65.8 |
| Diffprompt | **79.6** | **58.9** | **75.8** | 30.1 | **87.7** | **71.5** | **67.3** |

Following the training procedure described in Section 3.2, we present results on DomainNet in Table 4. As a discriminative model, the hypernetwork baseline fails to capture the distribution of collected prompts, leading to inferior performance compared to the diffusion-based method. Another alternative, "ERM for each", combines a domain classifier with domain-specific prompts trained via ERM. The classifier shares the same backbone as our condition module, with a classification head trained on the labeled set. At test time, it predicts the domain and selects the corresponding expert prompt. This baseline performs relatively well on the visually distinct Quickdraw domain but degrades on others due to misclassification errors. This comparison demonstrates the strength of diffusion-based prompt generation, which dynamically adapts to test-time inputs without requiring discrete domain labels.

**Computation resources.** Table 5 reports GPU memory, computation cost with a batch size of 64 on DomainNet and model size using PyTorch Profiler. The GPU memory and computation cost account for operations during adaptation and inference at test time. Our DiffPrompt requires lower resources because prompt generation does not involve gradient backpropagation, despite the 1000

Table 5: Computation resources.

| Method | Memory | Computation cost | Model size |
|---|---|---|---|
| TENT | 12.6 GB | 12.3 TFLOPs | 523.5 MB |
| CoTTA | 15.5 GB | 29.0 TFLOPs | 523.5 MB |
| RoTTA | 14.3 GB | 29.0 TFLOPs | 523.5 MB |
| RMT | 24.4 GB | 45.3 TFLOPs | 523.5 MB |
| DiffPrompt | 3.4 GB | 9.9 TFLOPs | 523.5+183.1 MB |

denoising steps in the diffusion model's inference. In contrast, the optimizer-based baselines require calculating the gradient of each CLIP parameter, and their total cost would continue to increase as CLIP scales up. However, a drawback of DiffPrompt is its model size, as it requires saving an extra diffusion model for prompt generation beyond CLIP.

## 5 CONCLUSION

We introduce Lifelong Continual Adaptation (LCA), a setting that models sequential and recurring in-distribution data streams, reflecting practical deployment scenarios overlooked by prior work. To address this challenge, we propose DiffPrompt, a diffusion-based framework that learns the distribution of domain-specific prompts and generates them conditioned on incoming data to adapt a frozen foundation model. By avoiding online parameter updates, DiffPrompt achieves stable and resource-efficient adaptation. Experiments on DomainNet and ImageNet-C show that DiffPrompt consistently outperforms ERM and continual TTA baselines, validating both the effectiveness of our approach and the importance of LCA as a realistic and non-trivial setting for continual adaptation.

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

## A    LIMITATIONS

LCA assumes that machine learning models are deployed in predictable recurring domains, such as different weather conditions or day–night cycles. However, real-world scenarios may also include outlier situations, such as rare geographic locations or extreme weather events. In such cases, a robust system should detect and report the anomaly rather than attempt erroneous adaptation. The current DiffPrompt framework does not explicitly address this issue. Future work may incorporate OOD detection mechanisms to complement LCA, enabling systems to handle both recurring domains and unexpected out-of-distribution inputs.

## B    DETAILS AND SETUPS OF BASELINES

**Zero-shot**    Zero-shot means that we use the base model, a pre-trained CLIP model, to be directly evaluated on the test data streams. The prompt applied to the text encoder is the commonly used template "a photo of a [CLASS]".

**ERM**    In this baseline, the CLIP model is trained with the training sets of all domains. The textual prompt is trainable while the CLIP model itself is frozen, as in CoOp Zhou et al. (2022b). The initialization of the prompt is the template "a photo of a [CLASS]". The training recipe is consistent with that in the prompt collection stage of our method. After training, the model with the trained prompt is evaluated on the test streams.

**TENT**    TENT Wang et al. (2021) employs an unsupervised loss—entropy minimization—to optimize the model using each batch of test data. This approach enhances prediction confidence on the test data, enabling the model to adapt to the incoming data stream.

**TPT**    TPT Shu et al. (2022) is a test-time prompt tuning method for CLIP. Instead of performing continual adaptation, it adapts the textual prompt for each test sample individually. Specifically, TPT augments the test image into a batch and optimizes the textual prompt with the batch using entropy minimization.

**SAR**    SAR Niu et al. (2023) introduces a selective and sharpness-aware entropy minimization loss, aiming to improve the reliability of online optimization using entropy minimization.

**CoTTA**    CoTTA Wang et al. (2022) uses a teacher-student self-training framework, where the student model is continually trained with the data in streams and pseudo labels from the teacher model. The teacher model is updated by an exponential moving average of the student weights, and it produces the pseudo labels with test-time augmented input data. In the continual test-time adaptation setting, the method continually optimizes a source model in the streams of sequential target domains. In our experiments, for consistency with other baselines and our method, this method is evaluated starting from the ERM-trained model in streams.

**RMT**    RMT Döbler et al. (2023) adopts a similar teacher-student self-training framework as CoTTA while further introducing a contrastive learning method with computed source prototypes and a source replay strategy. In our experiments, we also use the ERM-trained model as the initialization for this method. We use the training sets to compute the prototypes and conduct the source replay. Due to memory constraints, we do not include the evaluation of this method on the ImageNet-C dataset (1000 classes) because it needs to simultaneously keep two computational graphs for backpropagation for two losses associated with two different inputs.

**RoTTA**    RoTTA Yuan et al. (2023) is also based on a teacher-student self-training framework while it introduces a resampling approach to enhance stability and performance in online streams. We also evaluate the method using the ERM-trained base model.

## C  ADDITIONAL ABLATION STUDIES

**Number of Prompts.**   To analyze the sensitivity to the number of prompts used for diffusion training, we conduct ablation experiments with 20, 40, and 80 prompts (using the same random seed). As shown in Table 6, the results with fewer prompts are slightly worse than those with 80, but the difference is small. Since the number of domains is relatively low (6), even 20 prompts per domain are sufficient to capture a reasonably diverse prompt distribution.

Table 6: Sensitivity analysis on the number of prompts used during diffusion training.

| # Prompts | 20 | 40 | 80 |
|---|---|---|---|
| Accuracy | 67.2 | 67.1 | 67.3 |

## D  LLM USAGE

We used large language models (LLMs) only to aid in the polishing of writing. LLMs were employed to improve grammar, clarity, and readability of the manuscript. No LLM was used for research ideation, experimental design, or result generation.

