# OpenReview forum: "Generative Prompting with Diffusion for Lifelong Continual Adaptation"
_ICLR.cc/2026/Conference — ICLR 2026 Conference Withdrawn Submission_

### Official Review · Reviewer_PRaA · 2025-10-29

**Soundness:** 3
**Presentation:** 2
**Contribution:** 2
**Rating:** 2
**Confidence:** 4

**Summary:**

The paper introduces a new setting called Lifelong Continual Adaptation (LCA), similar in spirit to continual test-time adaptation (continual TTA), where a model hast to learn throughout sequential test-time streams consisting of recurring domains. The idea is to mimic real-world deployment conditions, where distributions reappear over time rather than being entirely new (as in continual TTA). To address this, the authors propose DiffPrompt, a method that uses a conditional diffusion model to generate domain-specific prompts for a frozen vision-language foundation model (CLIP). The diffusion model is trained on prompt samples from multiple domains and later generates new prompts conditioned on batches of incoming data. The model adapts to the stream of tasks continually without requiring gradient updates or retraining. Experiments on DomainNet and ImageNet-C show that DiffPrompt outperforms ERM and continual TTA baselines.

**Strengths:**

1. The proposed LCA setting highlights a gap between continual test-time adaptation (which assumes unseen domains) and realistic deployment scenarios (where domains recur). This proposal is interesting and worth exploring.
2. By avoiding gradient-based updates during adaptation, the proposed DiffPrompt has a lower computational cost compared to traditional TTA approaches.

**Weaknesses:**

1. The conditioning mechanism depends on batches of test examples to compute statistics (mean and variance of CLIP features; Eq. 10). This violates the independence requirement of test-time evaluation: the model implicitly accesses correlations between examples. This makes the reported results methodologically invalid. This is my primary concern and justifies rejection by itself.

2. The two-stage process (prompt collection + diffusion training) and the use of an autoencoder add unnecessary complexity. Since prompts are small-dimensional vectors, compressing them with an autoencoder could be excessive. An ablation on the impact of the use of an autoencoder could be useful.

3. The first two ablation studies are not interesting and do not isolate core factors, as they are just checking whether the diffusion training and prompt training worked effectively (so that their loss is minimized). For what concerns the other two, they can also be improved, as the diffusion model is compared only to a hypernetwork baseline, but not against simpler conditional generators (e.g., VAEs, Normalizing flows, Gaussians).

4. Much of Section 3 reiterates diffusion math unrelated to the proposed method, as the authors largely restate definitions and properties from prior diffusion literature. The paper lacks theoretical clarity and over-relies on mechanical reproduction of diffusion formalism.

5. (minor) No adaptation is applied to TTA baselines to make them competitive with the proposed methodology: since LCA is proposed as a new setting, baselines specific to it are missing. Existing TTA methods are designed for unseen domains, not recurring ones.

6. Writing (minor):

- Lines 47–48 and 126: the citations for catastrophic forgetting and continual learning are incorrect. They should reference canonical works, such as:
  - McCloskey & Cohen (1989), Catastrophic interference in connectionist networks: The sequential learning problem.
  - French (1999), Catastrophic forgetting in connectionist networks.
  - Kirkpatrick et al. (2017), Overcoming catastrophic forgetting in neural networks.
- Lines 74–75: the citation for prompt tuning is incorrect. It should reference the foundational works:
  - Lester et al. (2021), The Power of Scale for Parameter-Efficient Prompt Tuning.
  - Li & Liang (2021), Prefix-Tuning: Optimizing Continuous Prompts for Generation.
  These are instead correctly cited in the Related Work section.
- Lines 85–86: the conceptual derivation of point (2) is unclear. The logical connection or theoretical justification leading to this point should be explicitly detailed or reformulated for clarity.
- Lines 421–422: replace "present" with "presents".

**Questions:**

None

---

### Official Review · Reviewer_NYfn · 2025-10-30

**Soundness:** 3
**Presentation:** 3
**Contribution:** 2
**Rating:** 2
**Confidence:** 4

**Summary:**

This paper introduces Lifelong Continual Adaptation (LCA), a new continual learning setting. The data distribution shifts of LCA are not entirely new but recurring and long-term. Traditional TTA methods, relying on online unsupervised training, prove to be dispensable, unstable, and inefficient for this LCA scenario due to error accumulation and costly backpropagation on large foundation models.
To address such a challenge, this work proposes DiffPrompt, a diffusion-based framework that generates domain-specific prompts to guide a frozen vision foundation model (e.g., CLIP). DiffPrompt's training involves two stages: first, collecting domain-specific prompt samples by fine-tuning prompts on individual domains, and then training a conditional latent diffusion model on these collected prompts, conditioned by features extracted from incoming data batches. During deployment, the trained diffusion model directly samples prompts based on current batch statistics, enabling stable and resource-efficient adaptation without online backpropagation. Experiments on DomainNet and ImageNet-C demonstrate that DiffPrompt consistently outperforms ERM and continual TTA baselines.

**Strengths:**

The paper introduces a novel setting, Lifelong Continual Adaptation (LCA), which addresses recurring domain shifts in real-world deployments.

The proposed DiffPrompt is a diffusion-based, resource-efficient solution for continual adaptation, which generates domain-specific prompts in the test time scheme.

The work is supported by comprehensive experiments and ablations, rigorously validating DiffPrompt's effectiveness against baselines and its advantages in the new LCA setting.

**Weaknesses:**

Unfair Comparison: The paper's comparison of adaptation costs is problematic because it primarily focuses on deployment-time adaptation costs (Table 5) and overlooks the substantial pre-deployment training investment required for DiffPrompt. In contrast, many baseline methods either do not require extensive source domain training or have significantly lower adaptation costs overall.


Questionable Novelty Claim for Diffusion-based Prompt Generation: The paper's claim, "Our work is the first to show that diffusion models can also be used to generate prompts for continual adaptation," is inaccurate. The use of diffusion models to generate parameters for continual Test-Time Adaptation (TTA) is not entirely novel, with existing works (e.g., "Continual Adaptation: Environment-Conditional Parameter Generation for Object Detection in Dynamic Scenario") and more broadly, parameter generation methods, already exploring similar ideas.

Outdated Baseline Methods: The selection of baseline methods for comparison is largely outdated. The discussion primarily references only a few recent (2024) works, potentially missing more contemporary and strong baselines in the field of continual adaptation and TTA.


Increased Model Complexity and Parameters: During test-time adaptation, the proposed method requires an additional external diffusion model, which significantly increases the total number of parameters compared to other baseline methods. This makes DiffPrompt less parameter-efficient.

**Questions:**

See Weaknesses.

**Details Of Ethics Concerns:**

No concerns.

---

### Official Review · Reviewer_Fo7e · 2025-11-01

**Soundness:** 2
**Presentation:** 2
**Contribution:** 2
**Rating:** 2
**Confidence:** 4

**Summary:**

This paper introduces Lifelong Continual Adaptation (LCA), a new problem setting where a model pre-trained on multiple seen domains adapts to a recurring test stream of those same domains. The authors argue that standard Continual TTA is ill-suited for this ID (in-distribution) scenario. The proposed method, DiffPrompt, is a generative framework that avoids online optimization by training a conditional diffusion model to generate domain-specific prompts based on test batch statistics.

**Strengths:**

The paper's primary strength is its novel formulation of the LCA problem. The generative, backpropagation-free approach to adaptation is a novel idea for this problem. The authors benchmark against the mixed-domain ERM baseline, and they demonstrate that standard TTA methods perform poorly in this setting, which validates their motivation for a new approach.

**Weaknesses:**

1. The paper's primary motivation for replacing TTA is that backpropagation is "expensive and impractical" and "inefficient" for large models. However, the proposed solution, DiffPrompt, involves a 183.1MB diffusion model that must run a 1000-step reverse diffusion process for every test batch (Sec 4.4, Table 5). This is a highly computation-heavy operation, and the claim that this is more "efficient" or "practical" for resource-constrained devices than a single backward pass is unconvincing and contradictory. The TFLOPS comparison in Table 5 does not seem to adequately represent the practical latency of 1000 sequential sampling steps.

2. The experimental comparison to TTA methods (TENT, COTTA, RMT, ROTTA) is a strawman. These methods are (a) designed for a different problem (OOD adaptation, which the authors explicitly state LCA is not) and (b) designed for pure vision models, not CLIP. A more meaningful comparison would be against state-of-the-art TTA methods designed for CLIP ([1-3]) was included (and is a good start), the comparison set remains incomplete and largely irrelevant, failing to prove that DiffPrompt is superior to other adaptation strategies for VL models. Even for TTA methods, more recent methods should be compared ([4-6]).

3. The paper fails to provide a strong justification for using a complex diffusion model over a simpler alternative. The ablation study (Table 4) compares against a "Hypernetwork" baseline that performs worse than ERM, suggesting it is poorly tuned. It does not answer the more relevant question: could a simple, lightweight MLP be trained to regress from batch statistics to a domain prompt and achieve comparable performance? The extreme complexity of a diffusion model does not seem justified.

4. The core mechanism of DiffPrompt—using the batch statistics concat[µ, σ] as a condition (though not new in TTA, see [4,6])—relies on critical unstated assumptions. First, it assumes that every test batch is composed of samples from a single domain. If a test batch were mixed, the resulting statistics would be a meaningless average of two modes, and the conditional generation would fail. Second, this reliance on batch statistics makes the method's performance inherently dependent on batch size. These statistics would be extremely noisy and unreliable with small batches, yet the paper provides no sensitivity analysis and reports all results at a fixed, large batch size of 64. These assumptions of pure, large batches are unrealistic for many real-world streams and limit the method's applicability.

[1] Ra-tta: Retrieval-augmented test-time adaptation for vision-language models, ICLR2025

[2] Efficient and context-aware label propagation for zero-/few-shot training-free adaptation of vision-language model, ICLR2025

[3] DynaPrompt: Dynamic Test-Time Prompt Tuning, ICLR2025

[4] DPCore: Dynamic prompt coreset for continual test-time adaptation, ICML2025

[5] Vida: Homeostatic visual domain adapter for continual test time adaptation, ICLR2024

[6] Test-Time Model Adaptation with Only Forward Passes, ICML2024

**Questions:**

1. For the ERM baseline, are the models trained on a mixture of all domain data, or are they trained sequentially with domain labels? If the latter, this assumes domain information is available during training, which is not always the case.

---

### Official Review · Reviewer_rzQD · 2025-11-09

**Soundness:** 2
**Presentation:** 3
**Contribution:** 3
**Rating:** 6
**Confidence:** 4

**Summary:**

This paper introduces a problem setting termed lifelong continual adaptation (LCA) where models must adapt to sequential domains that recur over time. The authors propose DiffPrompt, a diffusion-based framework that generates domain-specific prompts to adapt frozen vision foundation models (CLIP) without requiring online parameter updates. They empirically evaluate their method against continual TTA baselines and demonstrate performance gains on DomainNet and ImageNet-C.

**Strengths:**

- The method consistently shows performance gains on both benchmarks compared to the considered TTA approaches.
- The method reduces memory and computation cost compared to the considered TTA approaches.

**Weaknesses:**

- The proposed problem setting assumes all recurring domains are known and labeled in advance. However, in real world deployment settings, such as the examples given (e.g., autonomous driving systems), a more realistic assumption would be partly seen/partly unseen domains. There could also be new domains that are encountered. The motivation for this strong assumption can be strengthened.
- Limited empirical evaluation. The method is only tested for two benchmarks and gains on ImageNet-C are relatively modest. LCA also intersects with domain-incremental continual learning methods. Comparisons to some baselines in that literature would strengthen empirical evaluation too.
- Missing methods in related work section. LCA has overlap with domain-incremental continual learning methods that tries to leverage existing domain information such as [1]. More specifically, works such as [2] leverage diffusion models to route representations between experts (pre-existing domain classifiers), which is conceptually aligned with DiffPrompt in that it retrieves knowledge of an old domain during test time.
[1] S-prompts learning with pre-trained transformers: An occam's razor for domain incremental learning. [2] Expert Routing with Synthetic Data for Continual Learning.

**Questions:**

See weaknesses above.

---

### Note · Authors · 2025-12-08

I have read and agree with the venue's withdrawal policy on behalf of myself and my co-authors.